Influences of heel height on human postural stability and functional mobility between inexperienced and experienced high heel shoe wearers

Chen Yiyang 1
Li Jing Xian 2
Wang Lin wanglin@sus.edu.cn 1
1 School of Kinesiology, Shanghai University of Sport , Shanghai , China
2 School of Human Kinetics, University of Ottawa , Ottawa , Ontario , Canada
Keogh Justin
Electronic publication date: 2020 Oct 28
Publication date: 2020
Volume: 8
Electronic Location ID: e10239
Received 2020 Jul 16; Accepted 2020 Oct 4
Copyright: ©2020 Chen et al.
Copyright year: 2020
Copyright holder: Chen et al.
License: This is an open access article distributed under the terms of the Creative Commons Attribution License, which permits unrestricted use, distribution, reproduction and adaptation in any medium and for any purpose provided that it is properly attributed. For attribution, the original author(s), title, publication source (PeerJ) and either DOI or URL of the article must be cited.
License URL: https://creativecommons.org/licenses/by/4.0/

Keywords: High heel shoes, Heel height, Wearing experience, Postural stability, Functional mobility

Funding: The National Natural Science Foundation of China 11572202 11772201 31700815 The Postgraduate Foreign Academic Visiting Project from the Shanghai University of Sport stfx20190105 This work was supported by the National Natural Science Foundation of China (No. 11572202, 11772201, and 31700815) and the Postgraduate Foreign Academic Visiting Project from the Shanghai University of Sport (No. stfx20190105). The funders had no role in study design, data collection and analysis, decision to publish, or preparation of the manuscript.

==============================
Background

High heel shoes (HHS) can affect human postural control because elevated heel height (HH) may result in plantar flexed foot and limit ankle joint range of motion during walking. Effects of HH and HHS wearing experience on postural stability during self-initiated and externally triggered perturbations are less examined in the literature. Hence, the objective of the present study is to investigate the influences of HH on human postural stability during dynamic perturbations, perceived stability, and functional mobility between inexperienced and experienced HHS wearers.

Methods

A total of 41 female participants were recruited (21 inexperienced HHS wearers and 20 experienced HHS wearers). Sensory organization test (SOT), motor control test (MCT), and limits of stability (LOS) were conducted to measure participant’s postural stability by using computerized dynamic posturography. Functional reach test and timed up and go test were performed to measure functional mobility. The participants’ self-perceived stability was assessed by visual analog scale. Four pairs of shoes with different HH (i.e., 0.8, 3.9, 7.0, and 10.1 cm) were applied to participants randomly. Repeated measures analysis of variance was conducted to detect the effects of HH and HHS wearing experience on each variable.

Results

During self-initiated perturbations, equilibrium score remarkably decreased when wearing 10.1 cm compared with flat shoes and 3.9 cm HHS. The contribution of vision to postural stability was larger in 10.1 cm HHS than in flat shoes. The use of ankle strategy worsened when HH increased to 7 cm. Similarly, the directional control of the center of gravity (COG) decreased for 7 cm HHS in LOS. Experienced wearers showed significantly higher percentage of ankle strategy and COG directional control than novices. Under externally triggered perturbations, postural stability was substantially decreased when HH reached 3.9 cm in MCT. No significant difference was found in experienced wearers compared with novices in MCT. Experienced wearers exhibited considerably better functional mobility and perceived stability with increased HH.

Conclusions

The use of HHS may worsen dynamic postural control and functional mobility when HH increases to 3.9 cm. Although experienced HHS wearers exhibit higher proportion of ankle strategy and COG directional control, the experience may not influence overall human postural control. Sensory organization ability, ankle strategy and COG directional control might provide useful information in developing a safety system and prevent HHS wearers from falling.

Introduction

High heel shoes (HHS) have been widely used among women in several centuries; 37% to 69% of women wear HHS daily (American Podiatric Medical Association, 2014). HHS are featured with heel evaluation, rigid heel cap and curved plantar region, which interfere with natural foot motion (Cronin, 2014). A more plantar flexed and supinated foot position can alter the distribution of plantar pressure, affect muscle activities around ankle joints, and limit the range of motion (ROM) of the ankle during standing and walking (Ko et al., 2009; Luximon et al., 2015; Simonsen et al., 2012). A number of studies have documented that the effects of HHS are not localized to the ankle; instead, a “chain reaction” of kinematic effects travels up the lower limb and disturbs the displacement of the center of mass (COM) (Chien, Lu & Liu, 2013; Cronin, 2014; Schroeder & Hollander, 2018). These biomechanical alterations can decrease perceived stability, impair postural control, and increase the risks of falling among HHS wearers (Luximon et al., 2015; Wan, Yick & Yu, 2019). The rate of high heels-related injuries increased from 7.1% to 14.1% during the 11-year period from 2002 to 2012. Most of the injuries were sprains or strains occurred to either the ankle or foot body regions (Barnish & Barnish, 2009; Moore et al., 2015).

One of the risk factors on high heels-related injuries is decreased postural stability among HHS wearers (Wan, Yick & Yu, 2019). Postural control is the ability to stabilize and restore the body’s COM relative to the base of support (BOS) during self-initiated and externally triggered perturbations (Horak, 2006; Winter, 1995). To maintain postural stability, a complex motor skill based on the interaction of proprioceptive, visual, and vestibular system is utilized in this process (Mancini & Horak, 2010). Wearing HHS can cause biomechanical constrains and disturb human movement strategies through reduced BOS and elevated heel height (HH) (Chien, Lu & Liu, 2013). The HHS wearers tend to apply different movement strategies (e.g., ankle and hip strategy) to maintain the stability of the body’s equilibrium with regard to elevated HH during standing, walking, and dynamic perturbations.

A number of studies found that different HH can influence postural stability through interfering with the stabilization of COM with respect to the BOS. Different sensory and movement strategies are also involved in the process of postural control in HHS wearers. Recent studies have examined that HHS wearers had significantly worse standing balance starting at seven cm HH by analyzing the center of pressure (COP) magnitude in quiet stance and limits of stability test (LOS) (Choi & Cho, 2006; Gerber et al., 2012; Mika et al., 2016). During extrinsic perturbations, previous studies demonstrated that HHS can impair human balance (e.g., sinusoidal oscillations and waist pulling) (Choi & Cho, 2006; Sun et al., 2017). When HH increased to 10 cm, increased use of ankle strategy, slow center of gravity (COG) movement velocity, and decreased body equilibrium were observed with increased HH (Hapsari & Xiong, 2016; Truszczyńska et al., 2019). However, no difference in the interaction of sensory systems was found in postural control among HHS wearers with increased HH (Hapsari & Xiong, 2016). It will be worthwhile to detect how sensory systems interact during postural control, and to what extend can HH affect movement strategy and influence human overall postural control accordingly.

HHS experience might be another vital factor that can influence HHS wearers’ postural stability as well. Previous research has shown significant muscular alterations, such as overwork muscle activities in medial gastrocnemius and peroneus longus, shortened calf muscles, and increased Achilles tendon stiffness after long-term use of HHS (Cronin, Barrett & Carty, 2012; Csapo et al., 2010; Kermani et al., 2018). These muscular accommodations around ankle joints can affect the efficient use of ankle strategies to return the body to equilibrium during standing (Chien, Lu & Liu, 2014; Rahimi et al., 2017; Wan, Yick & Yu, 2019). However, Xiong and Hapsari found no significant difference in self-initiated standing balance and functional mobility between experienced HHS wearers and inexperienced HHS wearers, although the experienced group showed higher directional control of COG in LOS (Hapsari & Xiong, 2016). Therefore, whether HHS wearing experience can influence human postural stability and functional mobility remains unclear.

Hence, the current study aims to investigate the effects of HH (i.e., 0.8, 3.9, 7.0, and 10.1 cm) and HHS experience on postural stability during dynamic perturbations, perceived stability, and functional mobility in women. We hypothesized that human postural stability could decrease with increasing HH, and HHS experience could improve performance in postural control and functional mobility test.

Materials & Methods

Participants

A total of 41 female participants were recruited from the local university and communities (21 inexperienced HHS wearers and 20 experienced HHS wearers). All participants had a shoe size of EU 36–39 and self-reported to be free from lower limb injuries for a minimum of six months prior to the study. Participants with any history of musculoskeletal, cardiovascular, neurological, and vestibular abnormalities were excluded from the experiment. Anthropometrics were measured prior to the experiment (i.e., body height, weight, foot length, and arch height). The measurements of foot length and arch height were taken under two conditions: 10% and 90% weightbearing loads (Zifchock et al., 2017). Arch height flexibility (AHF) was defined as the changes in arch height from 10% to 90% weightbearing conditions, normalized to 80% body weight. Experienced HHS wearers were those who had worn narrow-heeled shoes with a minimum HH of four cm more than twice per week and at least eight hours per day for one year. Inexperienced HHS wearers were participants wearing HHS less than once per week (Hapsari & Xiong, 2016; Wan, Yick & Yu, 2019). The study was approved by the ethics committee of Shanghai University of Sport (Number: 2018074), and all subjects were provided written consents prior to the experiment.

Experimental shoes

Experimental shoes with HH of 0.8, 3.9, 7.0, and 10.1 cm were used in the study (Fig. 1). All the experimental shoes were manufactured by the same manufacturer. The shoe style and materials were maintained the same to minimize confounding variance. Except for the 0.8 cm HHS as the baseline condition, the three other types of HHS were featured with narrow-heeled shoes (12.5 mm*12.0 mm). Participants were allowed to familiarize themselves with the most suitable experimental shoes with shoe size ranging from EU 36–39 prior to the experiment. The four HHS testing conditions were assigned to participants in random order.

Figure 1 (A) Size of the heel base and (B–E) experimental shoes with different HH.

Data collection

Postural control

NeuroCom Balance Manager System (Version 9.3, Natus Medical Incorporated, USA) SMART EquiTest was used to assess postural stability by measuring the participants’ COG alignment at a sampling frequency of 100 Hz after they were familiar with the experimental HHS (Chander et al., 2016; Hapsari & Xiong, 2016). Computerized dynamic posturography has been proven to be a “gold standard” for assessing postural stability with high reliability and validity (Harro & Garascia, 2019). Prior to the test, participants were secured with a protective vest from falling off the instrumentation. They were instructed to stand on the two force plates (23 cm*46 cm) with feet aligned with the platform axis as the initial position. SOT and LOS were used to test the participants’ standing balance during self-initiated perturbations, whereas postural stability during externally triggered perturbations was tested by motor control test (MCT). Participants were asked to stand still with their feet fixed in the initial position. A five-minute rest was allowed between three tests to prevent potential fatigue.

Sensory organization test (SOT).

SOT utilizes the sway-referencing capabilities of the visual surroundings and the support surface to evaluate the integration of the sensory systems in postural control by selectively disrupting somatosensory and/or visual information. Moderate to excellent reliability has been established in SOT among healthy adults (Ford-smit et al., 1995; Harro & Garascia, 2019; Tsang et al., 2004), and among patients with multiple sclerosis (Hebert & Manago, 2017)and transtibial amputation (Jayakaran, Johnson & Sullivan, 2011). The six testing conditions in SOT are described in Table 1 (Yin & Wang, 2020). Each testing condition was repeated three times. All the testing orders were randomly assigned to the participants (Dickin, 2010). The equilibrium and composite scores (0–100) represent the ability of the participants to maintain postural stability in each condition and overall postural control, respectively. The strategy scores (0–100) quantify the relative amount of movement about the ankle and hip strategies that participants used in maintaining postural stability. A strategy score approaching 100 indicates that ankle strategy is more dominant in maintaining balance, whereas a score closest to 0 suggests that the participant uses hip strategy dominantly to stabilize her body under each trial. Somatosensory (SOM), vestibular (VEST), and visual scores (VIS) (0–100) in sensory analysis quantify the participants’ ability to integrate proprioception, vestibulum, and vision information that contribute to balance, respectively.

Table 1 Six testing conditions of SOT.

Condition	Eyes	Support surface	Visual surroundings	Anticipated sensory systems	
1	Open	Fixed	Fixed	Somatosensory	
2	Closed	Fixed	Fixed	Somatosensory	
3	Open	Fixed	Sway referenced	Somatosensory	
4	Open	Sway referenced	Fixed	Vision and vestibular	
5	Closed	Sway referenced	Fixed	Vestibular	
6	Open	Sway referenced	Sway referenced	Vestibular	

Motor control test (MCT).

Postural stability under support surface perturbations was assessed by MCT (Fig. 2). The two force plates with translation capabilities in backward and forward directions can create six perturbing conditions which are small backward translation (SBT), medium backward translation (MBT), large backward translation (LBT), small forward translation (SFT), medium forward translation (MFT), and large forward translation (LFT). Each testing condition was repeated three times. The six testing conditions were assigned in random order. The displacement of the support surface is scaled to the participant’s height during each translation. The outcome measures were composite latency and amplitude scaling. Composite latency measures the reaction time from the initiation of translation of the platform to the displacement of COG in milliseconds. Amplitude scaling is measured for right leg in units of angular momentum and normalized to body height and weight, which quantifies the force generated from the lower limb in response to the external perturbations (Vanicek et al., 2013).

Figure 2 (A) Experimental setup and (B) the illustration of support surface translation in MCT.

Limits of stability test (LOS).

LOS quantifies the ability of participants to intentionally displace their COG within the BOS. In LOS, a computerized screen was placed in front of the participants. They were instructed to lean their body on the sagittal plane in each direction to reach to the target location displayed on the screen as quick as possible upon hearing an auditory cue. Then, participants were required to remain in that position for 10 s. The outcome measures were COG movement velocity and directional control (DCL). COG movement velocity in degree per second (°/s) represents the average COG movement speed from the initial place to the target position. Directional control was calculated as the amount of the COG movement toward the intended direction minus the amount of off-axis movement (Yin & Wang, 2020).

Functional mobility test

After postural control tests, functional reach test (FRT) and timed up and go test (TUGT) were performed to measure functional mobility. FRT measures the maximum forward reach of the participants. Participants were instructed to lean their body forward as far as possible without stepping or reaching for assistance. Three trials were conducted for data normalization purposes. In TUGT, participants were requested to sit on a standard chair with their back against the chair, arms resting on the chair’s arms. They were instructed to walk a 3 m straight line, make turns, walk back to the chair and sit down. Participants were asked to walk at their comfortable speed. The time between the participants’ buttocks leaving and touching the seat surface was recoded. The fastest among the three testing trials was used for data analysis (Schoppen et al., 1999).

Perceived stability

Thereafter, the participants were instructed to quantify their perceived stability in FRT on a visual analog scale (VAS). The scores range from 0–100. The VAS score of 0 indicates that the participants were perceived as unstable, whereas a score of 100 suggests the most stable situation that can be perceived.

Statistical analysis

All data were presented as mean ± standard deviation (SD). The normal distribution of data was examined by the Shapiro–Wilk test. Repeated measurement of ANOVA (HH * HHS wearing experience) was conducted to detect the effects of HH and HHS wearing experience on each variable. Simple main effect analysis was used for post hoc comparisons. Significance was set at an alpha level of p = 0.05. Partial eta-squared (η2) effect size, 95% confidence interval (CI), and F-statistic were reported. Statistical analysis was performed using SPSS 22.0 statistical software package (SPSS Inc., Chicago, USA).

Results

Demographic characteristics of the participants

Table 2 illustrates the characteristics of the participants. No significant differences were observed in age, height, weight, body mass index (BMI), foot length 10% weightbearing, foot length 90% weightbearing and AHF between the two groups. The experienced group showed significantly higher HHS wearing frequency than the inexperienced group (p < 0.001).

Table 2 Demographic data of the participants.

	Inexperienced HHS wearers (N = 21)	Experienced HHS wearers (N = 20)	
Age (years)	25.05 ± 1.63	23.05 ± 2.24	
Height (cm)	1.63 ± 0.05	1.63 ± 0.05	
Weight (Kg)	57.51 ± 7.87	56.33 ± 6.94	
BMI (Kg/m2)	21.62 ± 2.35	21.09 ± 2.62	
Foot length 10% weightbearing (mm)	231.43 ± 8.50	231.25 ± 9.50	
Foot length 90% weightbearing (mm)	234.95 ± 8.27	235.15 ± 10.25	
AHF (mm/kN)	0.90 ± 0.05	0.70 ± 0.04	
HHS wearing frequency (hours/week)	2.19 ± 4.61	28.33 ± 10.13*	
Notes.

BMI Body Mass Index

AHF Arch Height Flexibility

* inexperienced vs. experienced HHS wearers, p < 0.05.

Table 3 Comparison of outcome measures (means ± SD) in SOT for four HHS in inexperiencedand experienced groups.

	Inexperienced HHS wearers (N = 21)		Experienced HHS wearers (N = 20)	p values	
	0.8 cm	3.9 cm	7 cm	10.1 cm		0.8 cm	3.9 cm	7 cm	10.1 cm	Within groups	Between groups	Two-way interaction	
Equilibrium score												
C1	93.02 ± 3.72	93.40 ± 2.82	92.46 ± 3.36	91.52 ± 2.40		93.58 ± 2.54	93.57 ± 2.03	92.80 ± 2.61	91.37 ± 2.39	<0.001	0.735	0.877	
C2	90.76 ± 2.58	91.20 ± 4.21	89.44 ± 4.24	87.86 ± 4.25		91.37 ± 2.76	91.55 ± 2.59	90.83 ± 1.71	87.97 ± 4.65	<0.001	0.473	0.672	
C3	89.89 ± 4.42	89.91 ± 3.99	88.86 ± 4.59	86.52 ± 4.34		91.25 ± 3.21	91.08 ± 2.76	89.57 ± 4.01	87.95 ± 4.61	<0.001	0.226	0.921	
C4	85.35 ± 10.46	88.19 ± 9.99	87.78 ± 7.44	89.95 ± 3.40		88.93 ± 8.69	89.62 ± 6.08	90.00 ± 4.83	89.55 ± 4.78	0.187	0.340	0.425	
C5	80.11 ± 10.37	79.56 ± 9.39	81.14 ± 6.44	80.56 ± 4.78		81.82 ± 7.81	79.62 ± 9.01	80.78 ± 5.45	81.42 ± 6.27	0.563	0.763	0.799	
C6	72.97 ± 10.87	76.81 ± 9.46	77.25 ± 9.37	80.90 ± 5.23		76.70 ± 12.27	75.65 ± 9.80	79.10 ± 11.12	85.01 ± 4.80	<0.001	0.358	0.292	
COMP	83.52 ± 7.35	84.86 ± 5.94	84.86 ± 5.70	84.52 ± 3.16		85.85 ± 6.33	85.20 ± 4.70	85.95 ± 4.84	85.15 ± 5.09	0.776	0.463	0.533	
Strategy score												
C1	95.06 ± 2.47	84.86 ± 5.94	84.86 ± 5.70	84.52 ± 3.16		95.58 ± 1.34	95.90 ± 1.18	94.98 ± 1.73	94.13 ± 1.82	<0.001	0.318	0.900	
C2	93.90 ± 2.34	93.40 ± 2.82	92.46 ± 3.36	91.52 ± 2.40		94.78 ± 1.64	94.92 ± 1.57	93.92 ± 1.39	90.98 ± 3.55	<0.001	0.145	0.701	
C3	94.17 ± 2.01	91.20 ± 4.21	89.44 ± 4.24	87.86 ± 4.25		95.30 ± 1.11*	95.37 ± 1.11*	94.22 ± 2.03	93.17 ± 1.91*	<0.001	0.002	0.278	
C4	89.24 ± 3.40	89.91 ± 3.99	88.86 ± 4.59	86.52 ± 4.34		90.58 ± 2.44	90.70 ± 2.78	90.03 ± 2.30	89.73 ± 2.73	0.036	0.104	0.837	
C5	85.05 ± 4.37	88.19 ± 9.99	87.78 ± 7.44	89.95 ± 3.40		87.22 ± 2.76	85.50 ± 5.02	84.70 ± 3.42	84.68 ± 2.18*	<0.001	0.032	0.061	
C6	84.44 ± 4.33	79.56 ± 9.39	81.14 ± 6.44	80.56 ± 4.78		86.87 ± 3.55	87.10 ± 2.54*	85.27 ± 5.16	85.02 ± 4.80*	0.002	0.020	0.235	
Sensory analysis score												
SOM	97.86 ± 3.14	97.62 ± 3.32	96.81 ± 3.09	96.19 ± 3.93		97.90 ± 2.29	98.05 ± 1.82	98.05 ± 2.39	96.40 ± 3.95	0.031	0.450	0.756	
VIS	91.71 ± 9.56	94.48 ± 9.42	94.90 ± 6.06	98.57 ± 3.60		95.20 ± 9.05	95.95 ± 6.08	97.10 ± 4.12	97.95 ± 4.37	0.010	0.247	0.484	
VEST	86.10 ± 10.24	85.10 ± 9.08	87.62 ± 5.95	88.24 ± 4.89		87.45 ± 7.57	85.10 ± 9.57	87.15 ± 4.98	89.20 ± 6.70	0.097	0.781	0.872	
Notes.

SOM somatosensory score

VIS visual score

VEST vestibular score

* Inexperienced vs. experienced HHS wearers, p < 0.05.

SOT

The descriptive data of SOT are shown in Table 3. No statistically significant interaction was found between the HH and HHS wearing experience on the outcome measures of SOT (Table 3). The main effect of HH was significant for the equilibrium score in C1 (F(3, 38) = 8.342, p < 0.001, η2 = 0.202), C2 (F(3, 38) = 14.498, p < 0.001, η2 = 0.202), C3 (F(3, 38) = 10.428, p < 0.001, η2 = 0.202), and C5 (F(3, 38) = 10.920, p < 0.001, η2 = 0.202). No significant effect of HHS wearing experience was found on the equilibrium score. Post hoc analysis revealed significantly lower equilibrium score in 10.1 cm than seven cm HHS among experienced HHS wearers in C2 (p = 0.035, 95% CI [0.143–5.590]).

The main effect of HH was significant for the strategy score in six conditions (F(3, 38) = 12.234, p < 0.001, η2 = 0.176; F(3, 38) = 29.763, p < 0.001, η2 = 0.271; F(3, 38) = 21.591, p < 0.001, η2 = 0.356; F(3, 38) = 3.125, p = 0.036, η2 = 0.074; F(3, 38) = 10.598, p < 0.001, η2 = 0.214; F(3, 38) = 5.601, p = 0.002, η2 = 0.126). The main effect of wearing experience was also significant in C3 (F(1, 40) = 10.841, p = 0.002, η2 = 0.218), C5 (F(1, 40) = 4.977, p = 0.032, η2 = 0.022), and C6 (F(1, 40) = 5.857, p = 0.020, η2 = 0.132). The strategy score decreased significantly when HH increased to seven cm compared with flat shoes among experienced HHS wearers in C5 (p = 0.001, 95% CI=0.997–4.036). In C3, the experienced HHS wearers demonstrated significantly higher strategy score than inexperienced HHS wearers in flat shoes (t = −2.231, p = 0.033), 3.9 cm (t =  − 2.404, p = 0.023), and 10.1 cm HHS (t =  − 3.327, p = 0.002; Table 3).

Table 3 illustrates that the main effect of HH was significant for sensory analysis score in SOM (F(3, 38) = 3.059, p = 0.031, η2 = 0.099) and VIS (F(3, 38) = 4.270, p = 0.010, η2 = 0.099), but the main effect of wearing experience was undetected. Post hoc analysis showed that the sensory analysis score declined significantly in VIS when wearing 10.1 cm HHS compared with flat shoes in inexperienced wearers (p = 0.008, 95% CI [1.470–12.244]).

MCT

No significant interaction between the HH and wearing experience was detected on outcome measures of MCT. As shown in Table 4, the main effect of HH was significant for the composite latency (F(3, 38) = 3.121, p = 0.044, η2 = 0.080), whereas no significant difference was detected in the pairwise comparison. The HH revealed a significant main effect on amplitude scaling in SBT (F(3, 38) = 7.004, p < 0.001, η2 = 0.163), MBT (F(3, 38) = 3.630, p = 0.015, η2 = 0.092), SFT (F(3, 38) = 15.604, p < 0.001, η2 = 0.302), MFT (F(3, 38) = 24.919, p < 0.001, η2 = 0.409), and LFT (F(3, 38) = 9.522, p < 0.001, η2 = 0.209). No significant main effect was investigated for HHS wearing experience on amplitude scaling in six perturbing conditions. In MFT, the amplitude scaling was significantly higher when HH increased to seven cm compared with flat shoes among experienced wearers (p = 0.013, 95% CI [−2.193–0.207]).

Table 4 Comparison of outcome measures (means ± SD) in MCT for four HHS in inexperiencedand experienced groups.

		Inexperienced HHS wearers (N = 21)	Experienced HHS wearers (N = 20)	p values	
		0.8 cm	3.9 cm	7 cm	10.1 cm	0.8 cm	3.9 cm	7 cm	10.1 cm	Within groups	Between groups	Two-way interaction	
Latency COMP (milliseconds)										
		128.39 ± 8.27	128.39 ± 7.82	126.50 ± 5.29	126.39 ± 4.98	131.90 ± 11.14	134.25 ± 15.11	129.20 ± 9.02	128.80 ± 7.30	0.044	0.146	0.576	
Amplitude scaling											
B	S	1.72 ± 1.02	2.28 ± 1.49	2.44 ± 1.46	2.83 ± 1.15	1.80 ± 1.47	1.75 ± 1.12	2.35 ± 1.63	3.00 ± 1.56	<0.001	0.759	0.579	
	M	4.06 ± 2.24	4.61 ± 2.89	5.06 ± 2.36	4.61 ± 1.88	3.20 ± 1.94	4.10 ± 2.29	4.40 ± 2.09	4.45 ± 2.24	0.015	0.359	0.798	
	L	6.33 ± 2.95	6.78 ± 3.34	6.67 ± 3.12	6.78 ± 2.44	4.90 ± 2.77	5.65 ± 3.00	5.65 ± 2.41	6.50 ± 2.31	0.082	0.359	0.798	
F	S	2.28 ± 1.71	2.78 ± 1.52	3.28 ± 1.74	3.89 ± 1.32	2.05 ± 1.15	1.90 ± 1.02	2.25 ± 1.41	3.45 ± 1.64	<0.001	0.089	0.314	
	M	4.39 ± 1.97	5.44 ± 1.85	5.78 ± 1.80	6.94 ± 1.86	3.70 ± 1.81	4.90 ± 2.29	5.15 ± 2.41	6.55 ± 2.80	<0.001	0.337	0.970	
	L	6.83 ± 2.33	7.44 ± 2.59	7.78 ± 0.56	8.44 ± 2.18	5.65 ± 2.76	7.10 ± 2.69	8.15 ± 3.10	8.20 ± 3.02	<0.001	0.622	0.328	
Notes.

COMP composite score

B backward

F forward

S small

M medium

L large

* Inexperienced vs. experienced HHS wearers, p < 0.05.

Table 5 Comparison of outcome measures (means ± SD) in LOS for four HHS in inexperiencedand experienced groups.

	Inexperienced HHS wearers (N = 21)		Experienced HHS wearers (N = 20)	p values	
	0.8 cm	3.9 cm	7 cm	10.1 cm		0.8 cm	3.9 cm	7 cm	10.1 cm	Within groups	Between groups	Two-way interaction	
COG movement velocity (∘/s)											
	4.86 ± 1.79	4.46 ± 1.44	4.34 ± 1.39	3.58 ± 1.16		85.85 ± 6.33	5.30 ± 1.60	5.20 ± 1.51	4.84 ± 1.45	<0.001	0.155	0.659	
Directional control (%)												
	82.33 ± 5.62	82.00 ± 6.83	77.67 ± 8.48	68.62 ± 9.74		84.45 ± 3.98	84.95 ± 3.20	80.20 ± 5.65	77.45 ± 6.53*	<0.001	0.029	<0.001	
Notes.

COG center of gravity

* Inexperienced vs. experienced HHS wearers, p < 0.05.

LOS

As shown in Table 5, no statistically significant interaction was found between the HH and HHS wearing experience on COG movement velocity, whereas the two-way interaction was significant on directional control (F(3, 38) = 7.790, p < 0.001, η2 = 0.166). The main effect of HH was significant for COG movement velocity (F(3, 38) = 20.770, p < 0.001, η2 = 0.347) and directional control (F(3, 38) = 75.478, p < 0.001, η2 = 0.659). The significant main effect of wearing experience was also determined for directional control (F(1, 40) = 5.114, p = 0.029, η2 = 0.116). The results of post hoc analysis showed that COG movement velocity decreased significantly when wearing 3.9 cm HHS compared with 10.1 cm HHS among experienced wearers (p = 0.001, 95% CI [0.310°/s–1.480°/s]). Experienced HHS wearers exhibited significantly higher COG directional control than inexperienced wearers when wearing 10.1 cm HHS (t=-3.391, p = 0.002).

Functional mobility

Table 6 illustrates that the two-way interaction (HH * wearing experience) was significant for FRT distance (F(3, 38) = 3.858, p = 0.016, η2 = 0.090) and TUGT time (F(3, 38) = 9.883, p <0.001, η2 = 0.202). The main effect of HH was significant for FRT distance (F(3, 38) = 94.859, p <0.001, η2 = 0.709) and TUGT time (F(3, 38) = 127.372, p < 0.001, η2 = 0.766). Significant main effect of wearing experience was also determined for FRT distance (F(1, 40) = 10.840, p = 0.002, η2 = 0.217) and TUGT time (F(1, 40) = 10.639, p = 0.0021, η2 = 0.214). With respect to the results of the pairwise comparison, generally, functional mobility decreased as HH increased. FRT distance was significantly shorter in 10.1 HHS than in flat shoes (p <0.001, 95% CI=3.170–8.973 cm), 3.9 cm (p <0.001, 95% CI [4.254–8.146] cm), and seven cm HHS (p < 0.001, 95% CI [2.675–6.225] cm) among experienced wearers. TUGT time showed a significant difference when wearing different HHS in experienced and inexperienced wearers. Experienced wearers performed longer FRT distance than inexperienced wearers in 3.9 cm (t =  − 2.714, p = 0.010), seven cm (t =  − 2.805, p = 0.003) and 10.1 cm HHS (t =  −4.524, p < 0.001). Similarly, TUGT time in experienced wearers was significantly shorter than inexperienced HHS wearers in 3.9 cm (t = 3.528, p = 0.010), 7 cm (t = 3.117, p = 0.003), and 10.1 cm HHS (t = 3.698, p = 0.001).

Table 6 Comparison of outcome measures (means ± SD) in functional mobility test andperceived stability for four HHS in inexperienced and experienced groups.

	Inexperienced HHS wearers (N = 21)		Experienced HHS wearers (N = 20)	p values	
	0.8 cm	3.9 cm	7 cm	10.1 cm		0.8 cm	3.9 cm	7 cm	10.1 cm	Within groups	Between groups	Two-way interaction	
FRT (cm)												
	33.71 ± 4.50	32.01 ± 4.86	30.48 ± 4.39	23.81 ± 3.85		36.43 ± 4.79	36.29 ± 5.25*	34.54 ± 4.87*	30.09 ± 5.00*	<0.001	0.002	0.016	
TUGT (s)												
	6.59 ± 1.27	7.24 ± 1.06	7.64 ± 1.38	68.62 ± 9.74		5.97 ± 0.74	6.24 ± 0.72*	6.53 ± 0.82*	7.35 ± 1.10*	<0.001	0.002	<0.001	
VAS (0-100)												
	74.71 ± 32.31	71.00 ± 23.63	58.48 ± 19.10	34.43 ± 24.05		80.40 ± 30.76	90.35 ± 8.16*	77.75 ± 13.44*	52.65 ± 19.51*	<0.001	0.001	0.351	
Notes.

FRT functional research test

TUGT time up and go test

VAS visual analog scale

* Inexperienced vs. experienced HHS wearers, p < 0.05.

Perceived stability

The main effect of HH (F(3, 38) = 26.911, p < 0.001, η2 = 0.415) and wearing experience (F(1, 40) = 11.517, p = 0.001, η2 = 0.027) was significant for perceived stability. No significant two-way interaction was detected on perceived stability. The perceived stability was decreased with increased HH. Specificity, the perceived stability reduced significantly in 7 cm HHS relative to flat shoes (p = 0.001, 95% CI [5.530–26.049]) and 3.9 cm HHS (p = 0.029, 95% CI [0.940–23.060]) among experienced wearers. The inexperienced wearers also perceived significantly decreased stability with increased HH similar to the experienced wearers (Table 6). The experienced wearers perceived significantly higher stability than inexperienced wearers in 3.9 cm (t =  − 3.538, p = 0.002), seven cm (t =  − 3.719, p = 0.001), and 10.1 cm HHS (t =  − 2.656, p = 0.011).

Discussion

The main purpose of the study is to evaluate the effects of HH and HHS wearing experience on human postural stability under dynamic perturbations. During self-initiated standing perturbations, HHS wearers exhibited decreased equilibrium and strategy scores in 10.1 cm HHS, compared with flat shoes and 3.9 and seven cm HHS. Vision played a vital role in the integration of the sensory systems in the postural control process with elevated HH. With respect to the control of the COG movement, the COG movement velocity and directional control declined in 10.1 cm HHS compared with flat shoes and 3.9 cm HHS. During external support surface perturbations, the postural latencies tended to delay with elevated HH. Amplitude scaling increased when HH increased to 3.9 cm compared with flat shoes. Similarly, impaired functional mobility can be detected in 3.9 cm HHS contrary to flat shoes. However, experienced HHS wearers did not show significant higher composite equilibrium scores than novices as the authors hypothesized. No difference in the somatosensory function and postural responses under external perturbations was found between the two groups. Experienced wearers utilized higher proportion of ankle strategy and COG directional control in maintaining postural stability. They perceived higher stability and performed better functional mobility than inexperienced HHS wearers.

In SOT, decreased equilibrium and strategy scores were found in 10.1 cm HHS, compared with flat shoes and 3.9 and seven cm shoes. The ability to integrate the sensory systems to maintain the stability of the body’s equilibrium was impaired in 10.1 HHS. HHS wearers intended to use a larger portion of vision than proprioception in the postural control process when wearing 10.1 cm HHS. However, the anticipatory postural reactions from proprioceptive receptors played a vital role in maintaining balance, especially in the absence of vision (Mika et al., 2016). In SOT, the elevated HH may simulate an unstable condition. The sensory condition is more challenged because the support surface and vision are sway referenced. Humans can increase sensory weighting to vestibular and vision information for postural orientation when surrounded by these sway-referenced vision and unstable surfaces (Horak, 2006). Our study demonstrated that hip strategy was adopted more than ankle strategy by HHS wearers with increased HH under interfered conditions. With the increase in HH, the distance of the ankle and hip joints from the line of gravity is reduced (Stefanyshyn et al., 2000). HHS wearers cannot exert torque at the ankles to rapidly move the body’s COM (Horak & Kuo, 2000; Wan, Yick & Yu, 2019). A higher percentage of hip strategy is used to generate a larger torque about the hip joint to realign the COG in response to higher HH (Vanicek et al., 2013). The early activation of the hip flexors may be involved in response to the translation of support surface (Horak & Kuo, 2000). Our study’s results are in line with Xiong’s study, in which the hip strategy was used because the ankle strategy failed to maintain balance when wearing HHS (Hapsari & Xiong, 2016). The ankle strategy is the first postural control strategy adopted by humans to counteract small perturbations of the COG. On the contrary, hip strategy is used in response to larger perturbations. Human often utilize the combination of ankle and hip strategies for postural correction under external perturbations. The proportion of the strategies that distributed in the postural correction is organized by central nervous system (CNS), based on somatosensory input (Shumway-Cook & Horak, 1986). Our study showed that the HHS wearing experience had no significant effect on the overall human postural control. Human postural control is considered a complex motor skill with respect to the support surface, visual environment, and cognitive process (Shumway-Cook & Horak, 1986). Experienced wearers were found to adapt to walking regularity more flexibly under cognitive load than HHS novices (Schaefer & Lindenberger, 2013). Significant different muscle efforts were exerted in HHS experts compared with novices (Stefanyshyn et al., 2000). Studies have shown that wearing experience can influence the ankle ROM and muscle strength. A more supinated position was found in HHS experts compared with novices. The higher ankle ROM of inversion and plantarflexion might affect efficient force conduction on foot arch and increase the risk of anterior talofibular ligament sprains in experienced HHS wearer (Ebbeling, Hamill & Crussemeyer, 1994; Kim, Lim & Yoon, 2013). The long-term adaptation of the supinated position in HHS experts would shorten the length of muscle fibers, decrease the amount of cross bridges of the muscle fibers and disturb the function of calf muscles ultimately (Timmins et al., 2016). The muscle performance of calf muscles might be affected on account of the increased concentric contraction power in ankle inversion. The increased mediolateral instability would induce the changes in power production owning to the habitual use of narrow heels in experts (Stefanyshyn et al., 2000). In addition, the muscle performance of calf muscles may be affected on account of the decreased plantarflexion torque and higher reduction on plantarflexion power in experienced HHS wearers (Farrag & Elsayed, 2016). Generally, HHS experience might further influence muscle activities and cognitive processing. However, the ability to integrate the sensory systems in postural control was not altered; this finding is supported by Xiong’s study (Hapsari & Xiong, 2016).

With regard to MCT, the amplitude scaling increased significantly when HH reach 3.9 cm. Although the composite latency was 4.06% lower in 10.1 HH than in 3.9 cm HH, no significantly delayed postural latency in response to external perturbations was found in our study. Similarly, previous studies have shown no significant difference in postural reaction time when wearing flip-flops, clog style Crocs, and Vibram Five-Fingers (Chander et al., 2016). Footwear design characteristics may influence human postural reaction because elevated HH can disturb the ROM of ankle joints and affect human postural control in response to forward translations accordingly. When HH reached 3.9 cm, the increased amplitude scaling suggested that HHS wearers may alter motor output strategies to maintain postural stability under perturbations. In the motor output process, the gastrocnemius medialis (GM), gastrocnemius lateralis (GL), tibialis anterior (TA), and vastus lateralis (VL) were found to exert more effort when wearing seven cm HHS compared to flat shoes (Hapsari & Xiong, 2016). The threshold of afferent discharge of muscle spindle was raised. The HHS wearers’ postural control can be affected for the somatosensory alternation around the ankle and foot (Gefen et al., 2002). However, no adverse effect on postural reaction was found even in 10.1 cm HHS. This finding suggested that the delay of latency was often associated with neurological disorders and anatomical constraints, other than the footwear design (Redfern et al., 2001). Previous studies demonstrated that HHS can impair human balance during other extrinsic perturbations (e.g., sinusoidal oscillations and waist pulling) (Choi & Cho, 2006; Sun et al., 2017). Sun et al. found that the COP displacement increased, and the COP trajectory transferred to the medial foot significantly during AP and ML perturbations when wearing 6.6 cm compared with 0.8 cm HH. However, the study did not control the shoe design and applied three types of HHS in the experiment (Sun et al., 2017). Choi and Cho compared human balance control of HHS wearers in barefoot and high-heeled posture when experiencing a waist-pull perturbation by quantifying the displacement and velocity of the COP. Results suggested that human balance control was approximately twice worse in HHS than barefoot, and the perturbation amplitude was not attributed to the participants’ body weight and height (Choi & Cho, 2006). Experienced HHS wearers exhibited no improvement in postural control under dynamic perturbations. They applied different muscle activation patterns compared with inexperienced wearers. Experienced wearers exerted significantly more muscle activities on GM and less muscular effort on VL, TA, and erector spinae than novices in SOT (Hapsari & Xiong, 2016). During HHS walking, substantial increases in muscle fascicle strains and muscle activation were found in experienced HHS wearers compared with barefoot walking during the stance phase (Cronin, Barrett & Carty, 2012). Experienced wearers may regulate the flexibility of the neuromuscular system to adapt to possible perturbations (e.g., walking and external perturbations) and can vary according to different HHs (Alkjær et al., 2012).

Our study investigated that the COG movement velocity and directional control in LOS significantly decreased in 10.1 cm compared with that in 3.9 cm HHS. Consistent with the previous study, when HH increased to 10 cm, slower COG movement velocity was observed in 10 cm than in four cm HH in LOS (Mika et al., 2016). The increased HH may induce the fear of falling in HHS wearers. The HHS wearers manifested slow COG movement velocity, declined COG excursions, and worst directional control, particularly in the forward and backward directions (Hapsari & Xiong, 2016). The experienced HHS wearers showed higher percentage of directional control in 10.1 cm HHS. It may be due to the motor learning effects in the experienced wearer, resulting in superior ankle strategy in maintaining postural stability (Schaefer & Lindenberger, 2013). Nonetheless, another study suggested that the increased muscular coactivation around the ankle joint could enhance joint stiffness during HHS walking. The walking balance may be improved through altered muscle activation patterns (Alkjær et al., 2012; Nielsen & Kagamihara, 1993). The effects of muscle activation patterns on the postural control process in LOS among HHS wearers remain unclear.

The functional mobility was impaired when HH reached 3.9 cm. A number of studies have shown that walking in HHS may affect neuromechanics and kinematics of the lower limbs when HH increased to four cm HH (Naik et al., 2017). When walking in 4 and 10 cm HHS compared with flat shoes, the postural stability may be decreased on the account of high joint stiffness evaluated by muscle pair synchronization around the knee joint (Pratihast et al., 2018). Accordingly, the TUGT completion time was longer for impaired postural stability and reduced perceived stability, consistent with previous findings (Arnadottir & Mercer, 2000). Our study found that the experienced HHS wearers had significantly shorter TUGT completion time and FRT distance than the novices. Long-time use of HHS has been suggested to shorten the gastrocnemius muscle fascicles and increase the Achilles tendon stiffness, thereby contributing to a restricted ankle ROM and reduced functional reach mobility (Csapo et al., 2010). Cronin et al. suggested that experienced HHS wearers may have increased muscle fascicle strains and lower limb muscle activation than inexperienced wearers during HHS walking. This finding indicates chronic adaptations in muscle–tendon structure related to HHS (Cronin, Barrett & Carty, 2012). The experienced wearers could apply altered movement strategies to increase effort on muscular control around the knee and ankle joints, so as to obtain postural stability during HHS walking. However, high muscle activities may contribute to muscle inefficiency and raised energy cost during walking, thereby leading to muscle strains, muscle fatigue, and pain (Cronin, 2014; Csapo et al., 2010; Ebbeling, Hamill & Crussemeyer, 1994).

Although we found better functional mobility and higher perceived stability in experienced HHS wearer, no significant increase in overall postural control was detected in long-time HHS users in SOT. In functional tests, important resources, such as biomechanical constraints (e.g., strength and limits of stability), cognitive processing (e.g., learning and attention), movement strategies (e.g., anticipatory and voluntary), and sensory strategies (e.g., sensory integration and reweighting), are required for postural control. Thus, the loss of somatosensory in the foot and higher sensory weighting in vision cannot completely predict the deficiencies in functional mobility because the function depends on the aforementioned resources likewise (Horak, 2006; Horak & Kuo, 2000). In terms of HH, we assume that the decreased perceived comfort and loss of joint position may lead to low perceived stability, compromising functional mobility accordingly (Hong et al., 2005; Lee & Hong, 2005).

The limitation of the study is that the results may not be extrapolated to all HHS populations from different ages and health statuses, considering that we only recruited healthy young females in our study. Besides, the neuromuscular mechanism of postural control in HHS wearers is still unknown. The effects of HH and long-term use of HHS on lower limb muscle activities, muscle coordination, and Hoffmann reflex need to be further studied to elucidate how CNS controls motor output in the postural control process. Furthermore, to provide evidence-based information for clinicians, more cohort studies can be conducted to explore the relationship between wearing experience and HHS-related injuries such as metatarsalgia and ankle sprain.

Conclusions

Perceived stability and functional mobility decreased when wearing HHS. The vision system had high weight in maintaining postural stability when HH increased to 10.1 cm. During dynamic perturbations, higher percentage of ankle strategies and motor control strategies was exhibited when wearing 3.9 cm HHS compared with flat shoes. In terms of HHS experience, experienced HHS wearers used higher proportion of ankle strategy and COG directional control in postural control than novices. In addition, experienced wearers perceived higher postural stability and showed better functional mobility. It is recommended that on evaluating the postural stability of HHS wearers, sensory organization ability, ankle strategy and COG directional control could be considered to be useful in developing a safety system and prevent HHS wearers from falling.

Supplemental Information

Supplemental Information 1 Raw data of sensory organization test

Click here for additional data file.

Supplemental Information 2 Raw data of motor control test

Click here for additional data file.

Supplemental Information 3 Raw data of limits of stability tast

Click here for additional data file.

Supplemental Information 4 Raw data of functional reach test and timed up and go test data

Click here for additional data file.

Supplemental Information 5 Raw data of visual analog scale

Click here for additional data file.

Supplemental Information 6 Demographic characteristics of the participants

Click here for additional data file.

The authors appreciate the kind participation of all the subjects.

Additional Information and Declarations

Competing Interests

Author Contributions

Human Ethics

Data Availability

The authors declare there are no competing interests.

Yiyang Chen conceived and designed the experiments, performed the experiments, analyzed the data, prepared figures and/or tables, authored or reviewed drafts of the paper, and approved the final draft.

Jing Xian Li and Lin Wang conceived and designed the experiments, analyzed the data, authored or reviewed drafts of the paper, and approved the final draft.

The following information was supplied relating to ethical approvals (i.e., approving body and any reference numbers):

The ethics committee of the Shanghai University of Sport approved this study (2018074).

The following information was supplied regarding data availability:

The raw data are available in the Supplementary Files.

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
