# Peer review of "Influences of heel height on human postural stability and functional mobility between inexperienced and experienced high heel shoe wearers"

_PeerJ, doi:10.7717/peerj.10239_

## Round 0.1 · original submission · Major Revisions

· Academic Editor

Major Revisions

The two reviewers and I see many positives with this study and the manuscript in which these results are described. However, there are also a number of ways in which the manuscript can be improved.

Reviewer 1 ·

Basic reporting

no comment

Experimental design

no comment

Validity of the findings

no comment

Additional comments

This study aimed to investigate the influence of heel height and high-heeled shoes wearing experience on the postural stability and functional mobility. Different heel height shoes were worn in experienced and inexperienced females to perform sensory organization test, motor control test and limits of stability test. Interesting findings were presented, however, there are still several points were not clearly clarified, and these issues should be addressed before a further recommendation could be made.

Specific comments:
1. Line 38, in the conclusions of Abstract, authors repeated the key findings of the current study, just wondering if any practical or clinical implications could be made from findings of this study.
2. Line 53-54, formatting issue, please check and revise throughout the manuscript.
3. Line 56-58, this statement is not clear, also grammar mistake, please correct.
4. Line 88-90, the statement is not clear, please rewrite.
5. Line 143-147, authors used 0-100 to define the strategy scores to quantify the hip and ankle strategy, any reference to support this scoring system, and what is the reliability and repeatability of this scoring system.
6. Line 152, the Motor control test (MCT), the test protocol is not clear from the current text description, kindly suggest adding the Figure to assist illustration of the test.
7. Line 156, how was the ‘amplitude’ scaled? By what parameters?
8. Line 198, authors presented the demographic information in the Table 2, please also include other anthropometrics, such as leg length, foot length and width, which might also affect the stability and postural control.
9. Line 286-287, it is an interesting statement here that experienced HHS wears did not show ‘better’ postural control than inexperienced wearers. Please elaborate more details on this point. And how to define the ‘better’ in postural control?
10. Line 299-301, authors mentioned more hip strategy was adopted than the ankle strategy. It is an interesting and key statement here, please discuss more on this point, as the current description is not clear.
11. Line 341-343, authors discussed that the balance control is not sensible to the body weight and height. Same as #8 comment, wondering the leg length, foot length and width may affect the postural control.
12. Line 408, how to define the ‘better’? please use academic expression.
13. Line 411, authors did not include any test or discussion on the falling risk with high heel shoes. Suggestions shall originate from the main content of the study. Please revise.

Reviewer 2 ·

Basic reporting

no comments

Experimental design

no comments

Validity of the findings

no comments

Additional comments

Please refer to the attched file for details.

Annotated reviews are not available for download in order to protect the identity of reviewers who chose to remain anonymous.

---

## Round 0.2 · accepted · Accept

· Academic Editor

Accept

I thank the authors for attending to all of the comments of the reviewers. I am therefore happy to recommend this paper be accepted for publication in PeerJ.

Reviewer 1 ·

Basic reporting

no comment

Experimental design

no comment

Validity of the findings

no comment

Additional comments

The authors have made a good revision, it is suitable to be accepted now.

Reviewer 2 ·

Basic reporting

no comment

Experimental design

no comment

Validity of the findings

no comment